# Olanzapine Modulate Lipid Metabolism and Adipose Tissue Accumulation via Hepatic Muscarinic M3 Receptor-Mediated Alk-Related Signaling

**DOI:** 10.3390/biomedicines12071403

**Published:** 2024-06-25

**Authors:** Yueqing Su, Chenyun Cao, Shiyan Chen, Jiamei Lian, Mei Han, Xuemei Liu, Chao Deng

**Affiliations:** 1Fujian Maternity and Child Health Hospital, College of Clinical Medicine for Obstetrics & Gynaecology and Paediatrics, Fujian Medical University, Fuzhou 350005, China; syq0506@fjmu.edu.cn; 2School of Medical, Indigenous and Health Sciences, and Molecular Horizons, University of Wollongong, Wollongong, NSW 2522, Australia; cshiyan@fjmu.edu.cn (S.C.); jlian@uow.edu.au (J.L.); mei.han9962@gmail.com (M.H.); 3Department of Brain Science, Faculty of Medicine, Imperial College London, London SW7 2BX, UK; chenyun.cao19@imperial.ac.uk; 4Department of Neurology, The First Affiliated Hospital of Fujian Medical University, Fuzhou 350004, China; 5School of Pharmaceutical Sciences, Southwest University, Chongqing 400715, China; liuxm@swu.edu.cn

**Keywords:** antipsychotics, anaplastic lymphoma kinase, muscarinic M3 receptors, liver, lipid metabolism, Cevimeline

## Abstract

Olanzapine is an atypical antipsychotic drug and a potent muscarinic M3 receptor (M3R) antagonist. Olanzapine has been reported to cause metabolic disorders, including dyslipidemia. Anaplastic lymphoma kinase (*Alk*), a tyrosine kinase receptor well known in the pathogenesis of cancer, has been recently identified as a key gene in the regulation of thinness via the regulation of adipose tissue lipolysis. This project aimed to investigate whether Olanzapine could modulate the hepatic *Alk* pathway and lipid metabolism via M3R. Female rats were treated with Olanzapine and/or Cevimeline (an M3R agonist) for 9 weeks. Lipid metabolism and hepatic *Alk* signaling were analyzed. Nine weeks’ treatment of Olanzapine caused metabolic disturbance including increased body mass index (BMI), fat mass accumulation, and abnormal lipid metabolism. Olanzapine treatment also led to an upregulation of *Chrm3*, *Alk*, and its regulator *Ptprz1*, and a downregulation of *Lmo4*, a transcriptional repressor of Alk in the liver. Moreover, there were positive correlations between Alk and *Chrm3*, *Alk* and *Ptprz1*, and a negative correlation between *Alk* and *Lmo4*. However, cotreatment with Cevimeline significantly reversed the lipid metabolic disturbance and adipose tissue accumulation, as well as the upregulation of the hepatic Alk signaling caused by Olanzapine. This study demonstrates evidence that Olanzapine may cause metabolic disturbance by modulating hepatic Alk signaling via M3R, which provides novel insight for modulating the hepatic *Alk* signaling and potential interventions for targeting metabolic disorders.

## 1. Introduction

Anaplastic lymphoma kinase (*Alk*), a member of the receptor tyrosine kinase family, is a membrane-bound tyrosine kinase. It is well established that aberrant structure or dysregulated expression of *Alk* contributes to various types of cancer pathology [1]. In addition to its role in cancers, other biological roles of *Alk*, such as involvement in the development and function of the nervous system, have also been discovered recently [2,3,4]. Previous studies have suggested that *Alk* is an important molecular mediator in the excitatory synaptic transmission onto nucleus accumbens shell medium spiny neurons expressed dopamine D1 receptors and ethanol consumption, suggesting *Alk* to be one important molecular mediator of this interaction, and a potential target for therapeutic treatment of alcohol use disorder [3].

In particular, recent genome-wide association studies (GWASs) have suggested that several *Alk* variants are associated with metabolic traits such as adiponectin concentration [5], lipid and glucose homeostasis [6], and body mass index (BMI) [7,8]. Recently, Orthofer et al. reported that the sympathetic control of adipose tissue lipolysis was linked to the Alk pathway modulating energy expenditure [9]. This finding came under the spotlight [10], and was supported by a recent study which found that Augmentor α (Augα), one of newly discovered ligands of *Alk*, appeared to control body weight through a hypothalamic pathway [11]. Since the sympathetic system and the parasympathetic system are the two arms of the autonomic nervous system [12]. It has been reported that specific mutations of the Alk gene induce an imbalanced development of the autonomic system, which leads to neuroblastoma in some patients [13]. Meanwhile, recent evidence links parasympathetic regulation to metabolic abnormalities such as abnormal food intake, weight gain, obesity, and dyslipidemia [14,15,16]. It is rational to hypothesize that the parasympathetic system is involved in *Alk*-signaling-mediated metabolic modulation.

Acetylcholine is the major neurotransmitter of the vagus nerves, which innervate the liver [17,18]. It exerts parasympathetic actions by activating the acetylcholine (Ach) muscarinic M1–M5 receptors (M1R–M5R), a group of heterotrimeric G protein-coupled receptors (GPCRs) [19]. Apart from the central nervous system, M3Rs are broadly expressed in the non-neural tissues, including the hepatocytes [20,21]. M3R plays a key role in regulating insulin secretion and other metabolic functions [22].

Second-generation antipsychotics (SGAs), particularly Olanzapine and Clozapine, cause the highest incidence of metabolic side effects [23,24]. In addition to the increasing appetite stimulation and weight gain by blockade of hypothalamic 5HT2C and histamine H1 receptors, it is of general consensus that the mechanisms underlying SGAs-induced metabolic disturbance are likely to be multifactorial and involved in both peripheral and central mechanisms [25,26,27,28,29]. Our and other studies have identified that Olanzapine has a high binding affinity to the M3R and can act as a potent M3R antagonist to directly induce dyslipidemia, but the consequence is weight gain [30,31,32]. Cotreatment of Cevimeline, an M3R agonist, can attenuate Olanzapine-induced dyslipidemia in animal models [33,34]. These findings have suggested that the ACh muscarinic M3R may be a prime mechanism in SGA-induced metabolic dysregulation, and a potential candidature of possible therapeutic targets. However, the underlying mechanism is largely unknown. Therefore, this project aimed to further investigate whether the *Alk* pathway is associated with Olanzapine-induced metabolic disturbances via hepatic M3R-mediated parasympathetic regulations. 

## 2. Materials and Methods

### 2.1. Animal Experimental Design

As shown in the animal treatment schedule (Appendix A), female Sprague Dawley rats (200–230 g) purchased from the Animal Resource Centre (Perth, WA, Australia), were housed in pairs in controlled conditions (12/12 light/dark cycle, 22 °C) with ad libitum access to a standard chow diet (3.9 kcal/g; 10% fat, 74% carbohydrate, and 16% protein) and water. Following one week of acclimatization and one week of training in taking cookie dough pellets (30.9% cornstarch, 30.9% sucrose, 6.3% gelatin, 15.5% casein, 6.4% fiber, 8.4% minerals, and 1.6% vitamins), rats were assigned randomly into various treatment groups and housed individually throughout the experimental period. All of the experimental processes were approved by the Animal Ethics Committee, University of Wollongong (AE12/26 & AE15/03), complying with the Australian Code of Practice for the Care and Use of Animals for Scientific purposes. Based on the data from our previous studies, a power analysis (with type I error of 0.05 and 80% power) was conducted to determine sample sizes [34,35,36,37]. It was confirmed in our previous studies that sample sizes (n = 12/group for metabolic parameters and n = 6/group for mRNA expressions) have enough power to detect the changes induced by antipsychotic intervention in rats [34,36,37,38,39].

Experiment 1. Female rats (n = 12/group) were treated orally with Olanzapine (Zyprexa, Eli Lilly, Indianapolis, USA; 6 mg/kg/day mixed with 0.6 g cookie dough, bis in die (b.i.d.)) or Vehicle (0.6 g cookie dough without drugs, b.i.d.; control) for 9 weeks, as reported previously [37], to examine whether Olanzapine induced metabolic disorders through modulating the hepatic *Alk* and related signaling. 

Experiment 2. To further examine the M3R’s role in lipid metabolism associated with the hepatic *Alk* pathway by Olanzapine, this experiment tested the effects of a potent M3R agonist Cevimeline [40,41]. Female rats (n = 12/group) were treated with Olanzapine (Zyprexa, Eli Lilly, Indianapolis, IN, USA; 6 mg/kg/day mixed with 0.6 g cookie dough, ter in die (t.i.d.)), Cevimeline (Evoxac^®^ Daiichi-Sankyo, Taiwan; 27 mg/kg/day mixed with 0.6 g cookie dough, t.i.d.), or Olanzapine + Cevimeline (OLZ + CEV; 6 mg/kg/day + 27 mg/kg/day mixed with 0.6 g cookie dough; t.i.d.), or vehicle (0.6 g cookie dough without drugs, control) for 9 weeks, respectively. 

### 2.2. Tissue Collection

All animals were euthanized by carbon dioxide asphyxiation 2 h after the final treatment. The body weight and length of each individual animal were measured to calculate the BMI (body weight (g) divided by squared body length (cm)) [38,42]. The liver and white fat tissue (including inguinal, perirenal, periovary, and mesentery fat) were dissected and weighed. The adipose tissue index relative to body weight (total white fat tissue weight/body weight × 100) and the liver: body mass ratio (liver weight/body weight × 100) were calculated. Cardiac blood samples were collected into centrifuge tubes containing EDTA (1.6 mg/mL) and then centrifuged at 4000× *g* for 10 min at 4 °C. Both plasma and liver samples were frozen by liquid nitrogen immediately and then stored at −80 °C for further analysis.

### 2.3. Plasma Lipid Parameters Measurements 

Lipid parameters, including triglyceride (TG), total cholesterol level (TC), high-density lipoprotein cholesterol level (HDL-c), and low-density lipoprotein cholesterol level (LDL-c), were measured using Thermo Scientific Kits on a Konelab 30i biochemistry analyzer (Thermo Fisher Scientific Oy, Vantaa, Finland).

### 2.4. RNA Extraction and Gene Expression Analysis by Real Time PCR

Total mRNA from 15 mg liver tissue was extracted using a PureLink™ RNA Mini Kit (#12183025; Invitrogen Life Technologies, Carlsbad, CA, USA). A total of 20 μL cDNA was synthesized from 2 μg total RNA using a High Capacity cDNA Reverse Transcription Kit (# 4368814; Thermo Fisher Scientific, Waltham, MA, USA) at 25 °C for 10 min, 37 °C for 120 min, and 85 °C for 5 min. The mRNA levels were determined by qRT-PCR in duplicate on Quant Studio 5 Real-Time PCR Systems (Thermo Fisher, USA) using SYBR Green Master Mix (Life Technologies, Sydney, NSW, Australia) for *Alk* (forward: 5′-CAGCTATGCAGTAAACTTCC-3′; reverse: 5′-AATGGGTATCTTTCAGGGTC-3′), LIM-domain only 4 (*Lmo4*) (forward: 5′-CAACGTGTATCATCTCAAGTG-3′; reverse: 5′-AGTAGTGGATTGCTCTGAAG-3′), pleiotrophin (*Ptn*) (forward: 5′-ACCATGMGACTCAGAGATG-3′; reverse: 5′-GAACTGGTATTTGCACTCAG-3′), pleiotrophin/receptor protein-tyrosine phosphatase beta/zeta (*Ptprz1*) (forward: 5′-TTAGTCGTTTTGGAAAGCAG-3′; reverse: 5′-ATGTAGTACTTGTCAGTGGAG-3′), which act as the repressor, ligands, and activator of *Alk,* respectively [43,44,45]. TaqMan^®^ Gene Expression Assays (Life Technologies, Sydney, NSW, Australia) were used for *Chrm3* (Rn00560986_s1), *Actin* (Rn00667869_m1), *Gapdh* (Rn01775763_gl), *Rplp0* (Rn03302271_gH), and *Hprt1* (Rn01527840_m1). The PCR were run at 95 °C for 10 min, followed by 40 cycles (95 °C 15 s, 60 °C 1 min). Target gene relative expression levels were normalized to the four endogenous control genes (*Actin*, *Gapdh*, *Rplp0,* and *Hprt1*) and calculated with the 2^−∆∆CT^ method.

### 2.5. Statistical Analysis 

All data were analyzed using SPSS software (version 21.0, IBM, Armonk, NY, USA). The outliers were identified and removed using Boxplot. The distribution of the data from all experiments was examined by the Shapiro–Wilk test. For data with normal distributions, group comparisons in Experiment 1 were analyzed by an unpaired two-tailed *t*-test or Welch’s *t*-test, while the data from Experiment 2 were analyzed by a two-way ANOVA, followed by post hoc Tukey *t*-tests. Pearson’s correlation test was employed to determine the correlation among the measurements. For the data that were not normally distributed, a nonparametric Mann–Whitney U-test was used for groups comparison, and Spearman test was applied for correlation analysis. Statistical significance was accepted when *p* < 0.05. The results were presented as the mean with SD, or with median and 95% CI for data with abnormal distribution. All figures in the article were generated by GraphPad Prism 7.04 (GraphPad Software Inc., San Diego, CA, USA).

## 3. Results

### 3.1. The Effect of Olanzapine on Lipid Metabolism and the Hepatic Alk Signaling

As shown in Figure 1, the Olanzapine rats had significantly higher BMI than the control group (0.60 ± 0.02 vs. 0.57 ± 0.03; t = 2.19, df = 21, *p* = 0.02; Figure 1A). Compared to the control, the Olanzapine rats showed a significantly higher adipose tissue index (6.05 (4.88–6.64) vs. 4.09 (2.59–5.24); U = 27, *p* = 0.01; Figure 1B), and liver: body mass ratio (2.86 ± 0.22 vs. 2.69 ± 0.15 ; t = 2.1, df = 21, *p* = 0.04) (Figure 1C). For lipid parameters in the plasma, there were significant increases in plasma TG (0.46 (0.37–0.51) vs. 0.25 (0.23–0.38); U = 21, *p* < 0.01) , TC (1.85 (1.65–2.14) vs. 1.65 (1.45–1.73); U = 32, *p* = 0.03), and LDL-c (0.24 ± 0.11 vs. 0.16 ± 0.07; t = 2.07, df = 21, *p* = 0.05) levels in the Olanzapine group (Figure 1D). These results suggested that a metabolic disturbance occurred after treatment with the M3R antagonist Olanzapine.

As shown in Figure 2A, the mRNA level of Chrm3 was significantly upregulated after chronic Olanzapine treatment (1.45 ± 0.41 vs. 1.00 ± 0.26; t = 2.23, df = 10, *p* = 0.04). The relative mRNA level of *Alk* was significantly higher in the Olanzapine-treated group than the control (2.49 ± 0.86 vs. 1.00 ± 0.42; t = 3.79, df = 10, *p* < 0.01; Figure 2B). Similarly, the relative mRNA expression of *Prptz1*, a regulator of *Alk* activity, was also significantly upregulated in the Olanzapine group (1.78 ± 0.66 vs. 1.00 ± 0.16; Welch’s t = 2.81, df = 5.58, *p* = 0.03; Figure 2C). On the contrary, chronic Olanzapine treatment led to a significant downregulation of *Lmo4*, which is a transcriptional repressor of *Alk* (0.65 ± 0.32 vs. 1.00 ± 0.14; t = 2.40, df = 10, *p* = 0.03; Figure 2D). However, no significant change in *Ptn*, an *Alk* ligand, was observed between the two groups (0.88 ± 0.20 vs. 1.00 ± 0.17; t = 1.24, df = 10, *p* = 0.12; Figure 2E). Moreover, there was a positive correlation between the mRNA levels of *Chrm3* and *Alk* (Pearson r = 0.60, *p* = 0.03; Appendix A), and a positive correlation between the mRNA levels of *Alk* and *Ptprz1* (Spearman r = 0.69, *p* = 0.01; Appendix A), whereas there was a negative correlation between the mRNA levels of *Alk* and *Lmo4* (Spearman r = −0.65, *p* = 0.02; Appendix A).

Additionally, there were significant correlations between the gene expression of the *Alk* signaling and various lipid metabolic parameters, including *Alk* and TG (Pearson r = 0.87, *p* < 0.001; Appendix A), Alk and adipose tissue index (Pearson r = 0.59, *p* = 0.04, Appendix A), *Ptprz1* and TG (Spearman r = 0.71, *p* = 0.01; Appendix A), and *Ptprz1* and adipose tissue index (Spearman r = 0.63, *p* = 0.02; Appendix A), as well as Lmo4 and TG (Spearman r = −0.77, *p* < 0.01; Appendix A). These data provided further support that the lipid metabolic disturbance induced by M3R antagonist Olanzapine was associated with the activation of *Alk* signaling, which led to further investigation of the M3R agonist Cevimeline.

### 3.2. Cevimeline Cotreatment Ameliorates Olanzapine Effects on Lipid Metabolism and Related Hepatic Alk Pathway 

Two-way ANOVA analysis revealed that there were significant main effects of both Olanzapine (F_1,43_ = 6.556, *p* = 0.014) and Cevimeline (F_1,43_ = 27.10, *p* < 0.0001), as well as interactions between the two factors (F_1,43_ = 4.950, *p* = 0.031) on BMI value. Further Tukey’s multiple comparisons test showed that the BMI value was significantly increased in the Olanzapine group (vs. control, *p* < 0.05), but there was a significant decrease in the Cevimeline group (vs. control, *p* < 0.05), as well as in the OLZ + CEV group (vs. control, *p* < 0.05; vs. Olanzapine, *p* < 0.001; Figure 3A).

Meanwhile, significant main effects of both Olanzapine (F_1,42_ = 7.558, *p* = 0.009) and Cevimeline (F_1,42_ = 10.100, *p* = 0.003) were also observed on the adipose tissue index. The adipose tissue index was significantly increased in the rats with the Olanzapine treatment (vs. control, *p* < 0.01), but significantly decreased after the Cevimeline treatment (vs. control, *p* < 0.05) and OLZ + CEV cotreatment (vs. Olanzapine, *p* < 0.01; Figure 3B). 

As for liver: body mass ratio index, although there was not a significant main effect of either Olanzapine or Cevimeline, a significant interaction effect of them was observed when analyzed with two-way ANOVA (F_1,41_ = 8.508, *p* = 0.0057). The liver: body mass ratio index was significantly increased in the rats with the Olanzapine treatment (vs. control, *p* < 0.01), but significantly decreased after the OLZ + CEV cotreatment (vs. Olanzapine, *p* < 0.01; Figure 3C). 

The TG level significantly increased in the Olanzapine rats (vs. control, *p* < 0.01), but significantly decreased in the OLZ + CEV group (vs. Olanzapine, *p* < 0.01; Figure 3D). Meanwhile, TC concentration was significantly lower in the rats that received only Cevimeline treatment and the rats that received OLZ + CEV treatment (both vs. control, *p* < 0.01; Figure 3D). 

As shown in Figure 4, there were significant main effects of both Olanzapine (F_1,20_ = 6.361, *p* = 0.02) and Cevimeline (F_1,20_ = 9.152, *p* = 0.007), but not their interaction effect (F_1,20_ = 2.850, *p* = 0.106) on the mRNA of *Chrm3*. Post hoc analysis showed that the *Chrm3* level was significantly upregulated in the Olanzapine (vs. control, *p* < 0.01), but downregulated in the OLZ + CEV group (vs. Olanzapine, *p* < 0.01; Figure 4A). Although there was not a significant effect of Olanzapine (F_1,20_ = 0.0006, *p* = 0.98), a significant main effect of Cevimeline (F_1,20_ = 8.572, *p* = 0.008) and an interaction effect between the factors (F_1,20_ = 11.350, *p* = 0.003) were observed. *Alk* mRNA expression showed a significant upregulation in the Olanzapine group (*p* < 0.01) and a downregulation in the OLZ + CEV (vs. control, *p* < 0.05; vs. Olanzapine, *p* < 0.01; Figure 4B).

Similarly, there was not a significant main effect of Olanzapine (F_1,20_ = 3.501, *p* = 0.076), but there was a significant main effect of Cevimeline (F_1,20_ = 21.14, *p* = 0.0002), as well as an interaction between them (F_1,20_ = 12.31, *p* = 0.002) on the gene expression of *Prptz1*. The mRNA level of *Prptz1* was significantly higher in the Olanzapine rats than the control (*p* < 0.05), whereas a significant reduction in those receiving OLZ + CEV treatment was observed (vs. control, *p* < 0.05; vs. Olanzapine, *p* < 0.01; Figure 4C). For *Lmo4*, a significant Olanzapine main effect (F_1,20_ = 4.403, *p* = 0.049) was observed, and there was a significant decrease in *Lmo4* mRNA level in the Olanzapine-treated rats (*p* < 0.05; Figure 4D). There was no difference in the Ptn expression between the four groups (all *p* > 0.05; Figure 4E). Consistent with Experiment 1, there were positive correlations between *Chrm3* and *Alk* (r = 0.644, *p* < 0.001; Appendix A) and between *Alk* and *Ptprz1* (r = 0.557, *p* < 0.01; Appendix A), but a negative correlation between *Alk* and *Lmo4* (r = −0.474, *p* < 0.05; Appendix A). Meanwhile, significant correlations between the gene expression of *Alk* signaling and metabolic changes were also observed in the experiment with Cevimeline, including Chrm3 and TG (r = 0.429, *p* < 0.05; Appendix A), *Ptprz1* and TG (r = 0.513, *p* < 0.01; Appendix A), and *Ptprz1* and BMI (r = 0.477, *p* < 0.05; Appendix A). These results suggest that M3R agonist Cevimeline is able to partially reverse the metabolic disturbances caused by M3R antagonist Olanzapine through inhibiting the activation of the hepatic *Alk* signaling.

## 4. Discussion

This study expands the recent finding of a new physiological role for *Alk* in thinness through the central sympathetic control of adipose tissue lipolysis [9], and provides evidence that Olanzapine may cause metabolic disturbance by modulating hepatic *Alk* signaling via M3R. This evidence resulted from evaluating the effects of Olanzapine (an M3R antagonist) and/or Cevimeline (an M3R agonist) on hepatic transcriptional levels of genes associated with the *Alk* pathway and lipid metabolism in rats. A significant increase in the mRNA expression of *Alk* and its regulator *Ptprz1*, but a decrease in its transcriptional repressor *Lmo4* in the liver, were observed in Olanzapine treatment. This was consistent with the increases in BMI, adipose tissue index, liver: body mass ratio, and plasma lipid parameters induced by the chronic Olanzapine treatment. Cotreatment with Cevimeline reversed the fat accumulation and lipid metabolic disturbance caused by Olanzapine. In addition, the Cevimeline cotreatment inhibited the *Alk* signal pathway, which is characterized by the downregulation of *Alk* and *Ptprz1*. These findings suggest that the hepatic M3R regulates *Alk* modulation in lipid metabolism and adipose tissue accumulation. 

One previous study showed that stimulation of the vagus nerves reduces body weight and fat mass in rats [46]. Since acetylcholine is the major neurotransmitter of the vagus nerves and M3R is one of the major acetylcholine muscarinic receptors, it is not surprising in this study that the values of BMI, adipose tissue index, liver: body mass ratio, and lipid parameters were significantly increased after Olanzapine treatment, but decreased after cotreatment Cevimeline. More importantly, hepatic *Alk* mRNA expression was significantly activated by Olanzapine in rats, while the activated Alk signaling was inhibited when stimulated with an M3R agonist Cevimeline. It is worth noting that, although Olanzapine is a potent M3R antagonist, it also binds to other neurotransmission receptors, such as dopamine D2, 5-HT2C, and histamine H1 receptors [47]. The potential role of these receptors in the modulation of Olanzapine-induced activation of *Alk* signaling and related metabolic changes could not be completely excluded. However, Cevimeline is an M3R agonist with a high degree of M3R specificity and only a low M1 receptor affinity [40]. Since both *Alk* signaling activation and metabolic disturbance induced by Olanzapine were found to be attenuated after cotreatment with the Cevimeline in this study, this strongly supports the proposition that M3R regulates *Alk*-mediated metabolic traits. 

It is known that *Alk* transcription is repressed by *Lmo4* by interacting with estrogen receptorα (ERα) at the promoter region of *Alk* [43,48]. *Lmo4* is a protein containing two zinc-finger LIMs that interact with various DNA-binding transcription receptors [43]. Meanwhile, *Lmo4* was also reported to play a critical role in controlling central insulin signaling [49], and Lmo4 mutant mice show reduced energy expenditure, impaired leptin control of fat metabolism, and exhibited early onset adiposity [50]. Intriguingly, we observed a significantly lower expression of hepatic *Lmo4* after blocking M3R by Olanzapine in female rats, which resulted in weight gain, obesity, and lipid metabolic dysfunction. Moreover, a significant negative correlation between the hepatic expression of *Lmo4* and *Alk* was also found in this study. These observations suggest that the M3R-medicated parasympathetic control of *Alk* activation may act through inhibiting *Lmo4* expression, thus relieving the repression in the transcriptional control of the *Alk* gene. Meanwhile, a genome-wide association study reveals that *Ptprz1* plays a crucial role in Alk activation [45], and is strongly associated with feeding behavior traits [51]. Consistently, a significant positive correlation between hepatic *Alk* and *Ptprz1* expression was observed in this study. These results suggest that the Alk signaling-modulated metabolism is controlled via improving the transcription of its activator *Ptprz1*, and inhibits the mRNA expression of its repressor *Lmo4*.

Although the dosage of Olanzapine was 6 mg/kg/day in both Experiments 1 and 2, the daily dosage was delivered two times (3 mg/kg by 2 times) in Experiment 1 and three times (2 mg/kg by 3 times) in Experiment 2. It has been reported that both 2 mg/kg and 3 mg/kg Olanzapine in rats have a clinically comparable dopamine D2 receptor occupancy (of approximately 65–80%) [52,53]. In consideration of the half-life of Olanzapine being 24.2 and 72 h, respectively, in plasma and brain of humans [54], while 2.5 and 5.1 h in the plasma and brain (maintained at a high level for 8 h) of rats [55], the Olanzapine dosage in Experiments 1 and 2 should be equivalent and has a clinically comparable D2 receptor occupancy. The effects of only one dosage of Cevimeline were examined in this study, although these dosages have been proven to be effective in modulating lipid metabolism [33]. Further investigation with multiple dosages to confirm this finding is necessary. According to the dosage translation between species based on body surface area [56,57], the Cevimeline dosage used in this study may be higher than the clinical dosage [34,58]. However, the half-life of Cevimeline is shorter in rats (~2 h) than in humans (~5 ± 1 h) [58]. The current dosage has been reported without toxic effects in chronic studies up to 1 year, and without adverse effects on general behaviors, or nervous and other systems, although it may also act on other organs expressing muscarinic M3Rs [59,60,61]. 

It is worth noting there are several limitations in this study. One limitation is that only mRNA expression was examined in this study. Further studies are necessary to examine the protein levels with Western blot and immunohistochemistry to fully reveal the regulation of parasympathetic M3R on *Alk*-modulated lipid metabolism. In addition, it is also valuable in a future study to investigate the effects of M3R knockout on *Alk*-modulated lipid metabolism. In addition, only female rats were used in this study, because the female subjects were observed to be more sensitive to Olanzapine-induced metabolic disturbances in both humans and rats [62,63,64]. Indeed, a male rodent model of Olanzapine-induced metabolic disturbance could not be consistently constructed unless given a higher-fat diet instead of the standard low-fat chow [65]. This limitation would hinder our results from being transferred to a clinical setting; therefore, further research is necessary to examine the effects of Cevimeline intervention in preventing metabolic side effects of Olanzapine in a male cohort. Furthermore, it is worth noting that the variation within groups in both TC and HDL-c was very large; these variations may be explained by the fact that a certain proportion of samples were in the different level of hemolysis. A previous study suggested that hemolysis is the most common reason for interferences with some biochemistry parameters, including TC and HDL-c [66]. Another limitation is that the sample size is relatively small, although a power analysis was conducted based on the data from our previous studies. Therefore, further experiments in a large sample size will be valuable to confirm the findings in this study. 

In summary, in a rat model treated with parasympathetic M3R antagonist Olanzapine and/or an M3R agonist Cevimeline, this study found a number of metabolic disturbances and activation of the hepatic *Alk* pathway caused by Olanzapine, whereas they were attenuated when cotreated with an M3R agonist. In addition to Olanzapine, it has been reported that Clozapine also induces metabolic disorders through a muscarinic M3 receptor mechanism [32]. Further studies are important to investigate whether Clozapine-induced metabolic disorders are also mediated by *Alk* signaling, and can be attenuated by cotreated with Cevimeline. Following the recent identification of *Alk* as a key gene in the regulation of thinness, it has been proposed that *Alk* inhibition is a potential target for obesity therapy [9]. In fact, a number of *Alk* inhibitors (e.g., Crizotinib, Ceritinib, and Lorlatinib) have been used for treating some types of cancers for decades, but with a number of adverse effects [10,67,68,69]. This study provides novel evidence of a regulatory role of the hepatic M3R in *Alk*-modulated lipid metabolism and adipose tissue accumulation, which provides a new direction for the development of drugs for use in modulating hepatic *Alk* signaling that warrant further investigation.

## Figures and Tables

**Figure 1 biomedicines-12-01403-f001:**
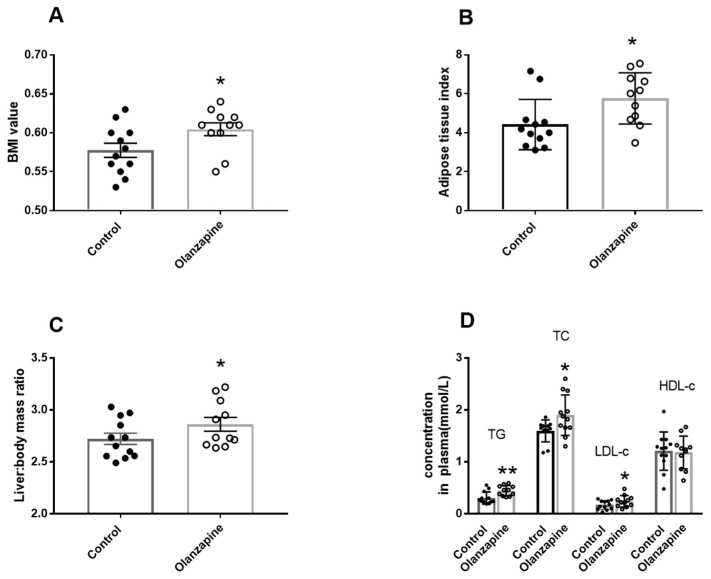
Profiles of metabolic changes in rats after treatment with Olanzapine. (**A**) BMI (body weight (g) divided by squared body length (cm)); (**B**) white adipose tissue index (adipose tissue weight/body weight × 100); (**C**) liver: body mass ratio (liver weight/body weight × 100); (**D**) lipid parameters in the plasma. Olanzapine: 6 mg/kg/day, 9 weeks. Data presented as mean ± SD (Olanzapine: n = 11, control: n = 12). * *p* ≤ 0.05, ** *p* < 0.01 vs. control. Abbreviations: BMI, body mass index; TG, triglyceride; TC, total cholesterol; HDL-c high-density lipoprotein cholesterol; LDL-c low-density lipoprotein cholesterol.

**Figure 2 biomedicines-12-01403-f002:**
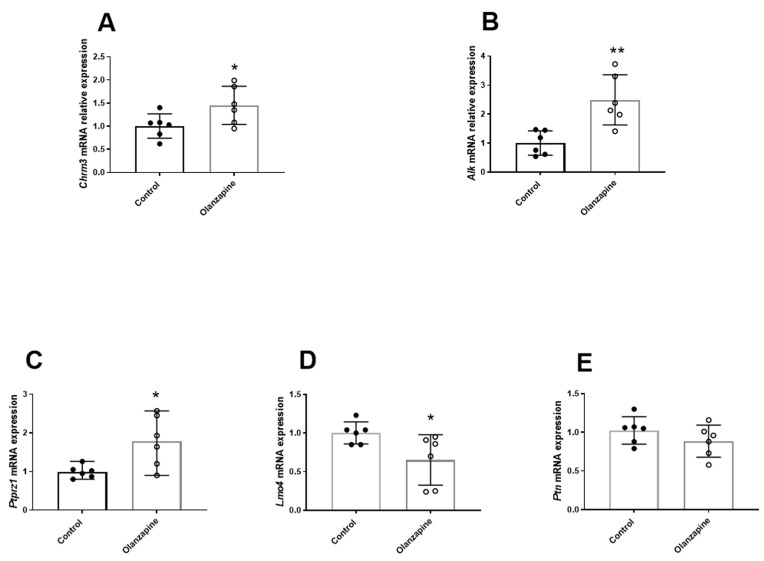
Effect of Olanzapine on gene expression of *Alk* and related pathways in the liver. (**A**) *Chrm3*, (**B**) *Alk*, (**C**) *Ptprz1*, (**D**) *Lmo4*, (**E**) *Ptn*. Data are presented as mean ± SD. The sample size is 6 per group. * *p* < 0.05, ** *p* < 0.01 vs. control.

**Figure 3 biomedicines-12-01403-f003:**
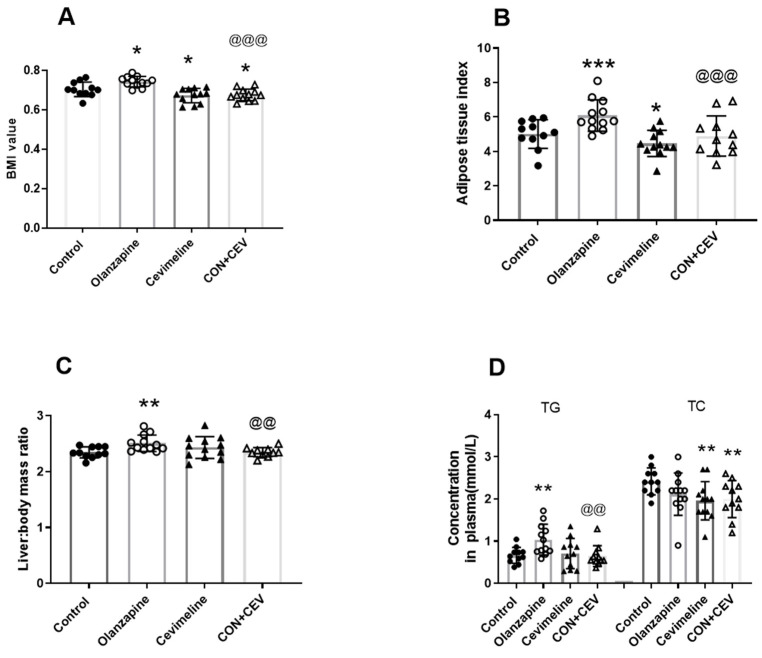
Profiles of metabolic changes after treatment with Olanzapine and/or Cevimeline. (**A**) BMI (body weight (g) divided by squared body length (cm)); (**B**) white adipose tissue index (adipose tissue weight/body weight × 100); (**C**) liver: body mass ratio (liver weight/body weight × 100); (**D**) lipid parameters in the plasma. Data presented as mean ± SD. The sample size is 12 per group. * *p* < 0.05, ** *p* < 0.01, *** *p* < 0.01 vs. control; @@ *p* < 0.01, @@@ *p* < 0.001 vs. Olanzapine. Abbreviations; BMI, body mass index; TG, triglyceride; TC, total cholesterol; HDL-c, high-density lipoprotein cholesterol; LDL-c, low-density lipoprotein cholesterol.

**Figure 4 biomedicines-12-01403-f004:**
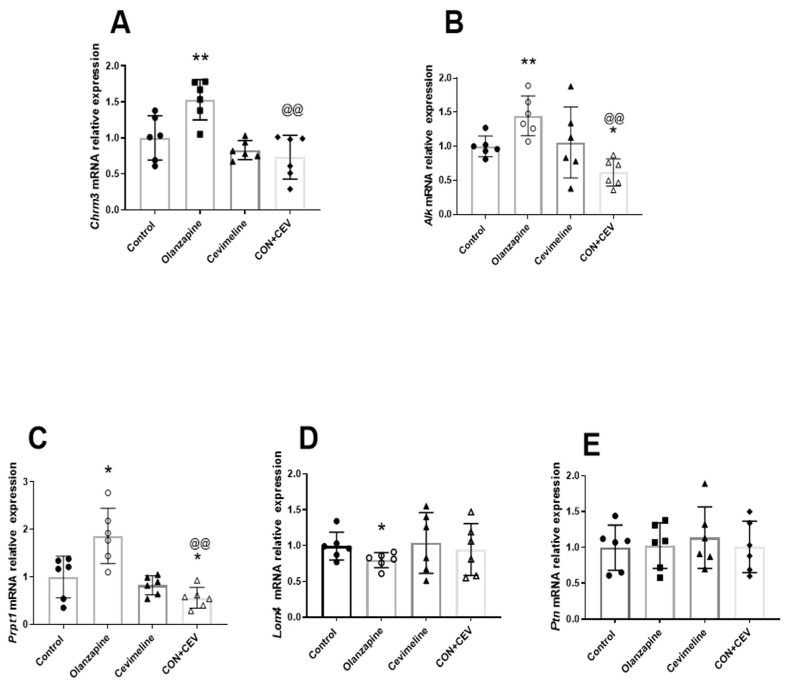
Effect of Olanzapine and/or Cevimeline on mRNA expression of *Alk* and related pathways in the liver. (**A**) *Chrm3*, (**B**) *Alk*, (**C**) *Ptprz1*, (**D**) *Lmo4*, (**E**) Ptn. Data are presented as mean ± SD. The sample size is 6 per group. * *p* < 0.05, ** *p* < 0.01 vs. control; @@ *p* < 0.01 vs. Olanzapine.

## Data Availability

The original contributions presented in the study are included in the article/supplementary material, further inquiries can be directed to the corresponding author.

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
