# Peer review of "Olanzapine Modulate Lipid Metabolism and Adipose Tissue Accumulation via Hepatic Muscarinic M3 Receptor-Mediated Alk-Related Signaling"

_biomedicines, 2024, doi:10.3390/biomedicines12071403_

Round 1

Reviewer 1 Report

Comments and Suggestions for Authors

The manuscript entitled, "Olanzapine Modulated hepatic Alk signaling and Lipid Metabolism via Acetylcholine Muscarinic M3 Receptors', is a nice piece of work that demonstrates the regulation of hepatic Alk signaling via a selected drug. However, the quality of the present work may be improved further by addressing the following queries:

1.      In the abstract section, repetition of terms such as Cevimeline (an M3R agonist) should be avoided.

2.      In the introduction section, authors should discuss why the selected drug(s) is chosen for the present study. Authors should also discuss the lists of other drugs working in the same fashion to broaden the scope of the present study.

3.      In the materials and methods section, authors should discuss the toxicity level/effects of the selected drugs if any and safer/recomended dose level of the drug

4.      In the result section pertaining to Figure 1(d), Authors should discuss why the trends of drug for HPL-c is not observed as compared to trends observed for other components as seen in Figure 1a-d.

5. In the figure 2d&2e, the effect of Olanzapine on the pattern of gene expression is quite different from the pattern reported in 2a-c. Authors should discuss the scope of these patterns in the potential repair/therapy system.

Author Response

  1. In the abstract section, repetition of terms such as Cevimeline (an M3R agonist) should be avoided.

Reply: Followed the comment, we have checked and deleted the repetitive description in the whole manuscript.

  1. In the introduction section, authors should discuss why the selected drug(s) is chosen for the present study. Authors should also discuss the lists of other drugs working in the same fashion to broaden the scope of the present study.

Reply: Followed the comments, we provided the justification why antipsychotic drug Olanzapine, as well as Cevimeline, were selected in this study. We also discussed and extended our findings to another antipsychotic drug Clozapine, as follows:

“Second generation antipsychotics (SGAs), particular olanzapine and clozapine, causes the highest incidence of metabolic side effects. Besides the increasing appetite stimulation and weight gain by blockade of hypothalamic 5HT2C and H1 receptors, it is a general consensus that the mechanisms underlying SGAs-induced metabolic disturbance are likely to be multi-factorial and involved in both peripheral and central mechanisms [23-27]. Our and other studies have identified that olanzapine has a high binding affinity to the M3R and can act as a potent M3R antagonists to directly induce dyslipidemia but the consequence of weight gain [28-30]. Co-treatment of Cevimeline, a M3R agonist, can attenuate olanzapine-induced dyslipidaemia in animal model [31, 32]. These findings suggested that the ACh muscarinic M3R may be a prime mechanism of SGA-induced metabolic dysregulation, and a potential candidature of possible therapeutic targets. However, the underlying mechanism is largely unknown …” (see Page 2, lines 64-75).

“Besides Olanzapine, it has been reported that Clozapine also induced metabolic disorders through a muscarinic M3 receptor mechanism [30]. Further studies are important to investigate whether Clozapine-induced metabolic disorders are also mediated by Alk signalling, and could be attenuated by co-treated with Cevimeline. “ (see Page 10, lines 361-365)

  1. In the materials and methods section, authors should discuss the toxicity level/effects of the selected drugs if any and safer/recomended dose level of the drug

Reply: Followed the comments, we discussed issues in dosages and the toxicity level/effects of the selected drugs, as follows:

“Although the dosage of Olanzapine was 6 mg/kg/day in both Experiments 1 and 2, the daily dosage was delivered in two times (3 mg/kg by 2 times) in Experiment 1, but three times (2 mg/kg by 3 times) in Experiment 2. It has been reported that both 2 mg/kg and 3 mg/kg Olanzapine in rats have a clinically comparable dopamine D2 receptor occupancy (of approximately 65–80%) [47, 48]. In consideration of the half-life of Olanzapine is 24.2 and 72 hours respectively in plasma and brain of humans [49], while 2.5 and 5.1 hours in the plasma and brain (maintained at a high level for 8 hours) of rats [50], the Olanzapine dosage in Experiments 1 and 2 should be equivalent and has a clinically comparable D2 receptor occupancy. The effects of only one dosage of Cevimeline have been examined in this study, although these dosages have been proven to be effective in modulating lipid metabolism [31]. Further investigation with multiple dosages to confirm this finding is necessary. According to the dosage translation between species based on body surface area [51, 52], the Cevimeline dosage used in this study may be higher than the clinical dosage [32, 53]. However, the half-life of Cevimeline is shorter in rats (~2 hours) than in hu-mans (~5 ± 1 hours) [53]. The current dosage has been reported without toxic effects in chronic studies up to 1 year, and without adverse effects on general behaviors, or nervous and other systems, although it may also act on other organs expressed muscarinic M3Rs [54-56].” (see Page 9, lines 323-340)

  1. In the result section pertaining to Figure 1(d), Authors should discuss why the trends of drug for HPL-c is not observed as compared to trends observed for other components as seen in Figure 1a-d.

Reply: Thank you for the comment. We agree that the variation within groups of both TC and HDL-c was quite large. We have discussed the limitation as follows:

“Furthermore, it is worthy to note that the variation within groups in both TC and HDL-c was very large, these variations may be explained that a certain proportion of samples were in the different level of hemolysis. A previous study has suggested that hemolysis is the most common reason in interferences with some biochemistry parameters, including TC and HDL-c [61].” (see Page 10, lines 353-357)

  1. In the figure 2d&2e, the effect of Olanzapine on the pattern of gene expression is quite different from the pattern reported in 2a-c. Authors should discuss the scope of these patterns in the potential repair/therapy system.

Reply: Thank you for the comment, we discussed this issue as follows:

“It is known that Alk transcription is repressed by Lmo4 by interacting with estrogen receptorα (ERα) at the promoter region of Alk [38, 43]. Lmo4 is a protein containing two zinc-finger LIMs that interact with various DNA-binding transcription receptors [38]. Meanwhile, Lmo4 was also reported to play a critical role in controlling central insulin signaling [44], and Lmo4 mutant mice show reduced energy expenditure, impaired leptin control of fat metabolism, and exhibit early onset adiposity [45]. Intriguingly, we observed a significant lower expression of hepatic Lmo4 after blocking M3R by Olanzapine in female rats, which resulted in weight gain, obesity and lipid metabolic dysfunction. Moreover, a significant negative correlation between the hepatic expression of Lmo4 and Alk was also found in this study. These observations suggest that the M3R-medicated parasympathetic control of Alk activation may act through inhibiting Lmo4 expression, thus relieving the repression in the transcriptional control of the Alk gene.”. (see Page 9, lines 305-316)

Reviewer 2 Report

Comments and Suggestions for Authors

The project aimed to investigate whether olanzapine modulated hepatic Alk pathway and lipid metabolism via M3R.

The abstract does not reflect the obtained results; the sentences are too general.

"Besides its role in cancers, other biological roles of Alk, such as involvement in the development and function of the nervous system, have also been discovered recently [2-4]." - too broadly stated. In the introduction, it is necessary to include what has been published so far and what is the novelty of this article. There are many articles on olanzapine and metabolic disorders.

 For such a small sample size, it is recommended to use non-parametric counterparts of statistical tests. This applies to both between-group differences and correlations. In this case, the applied parametric counterparts do not reflect the obtained results.

It is unclear in the results where and what test was used. It is recommended to report results according to standards, e.g., for the Mann-Whitney test: U = 23; p = 0.02.

Appropriate effect size measures were not calculated for the applied statistical tests. The p-value alone is definitely not enough. Descriptive statistics should be expanded to include key measures, such as the median and 95% CI.

The conclusions and discussion are not supported by a thoroughly conducted analysis. At this point, the article does not bring new findings.

Comments on the Quality of English Language

Extensive editing of English language required.

Author Response

  1. The abstract does not reflect the obtained results; the sentences are too general.

Reply: In abstract, we have already presented our findings as followed:

“Nine weeks’ treatment of Olanzapine caused metabolic disturbance including increased body mass index (BMI), fat mass accumulation, and abnormal lipid metabolism. Olanzapine treatment also led to an upregulation of Chrm3, Alk and its regulator Ptprz1, and a downregulation of Lmo4, a transcriptional repressor of Alk in the liver. Moreover, there were positive correlations between Alk and Chrm3, Alk and Ptprz1, and a negative correlation between Alk and Lmo4. However, co-treatment with Cevimeline significantly reversed the lipid metabolic disturbance and the up-regulation of the hepatic Alk signaling caused by Olanzapine. This study demonstrates evidence that Olanzapine may cause metabolic disturbance by modulating Alk signaling via M3R, which provides novel insight for modulating the hepatic Alk signaling and potential interventions for targeting metabolic disorders.” (See Page 1, lines 20-29)

  1. "Besides its role in cancers, other biological roles of Alk, such as involvement in the development and function of the nervous system, have also been discovered recently [2-4]." - too broadly stated. In the introduction, it is necessary to include what has been published so far and what is the novelty of this article. There are many articles on olanzapine and metabolic disorders.

Reply: Thanks for the comment. Followed the general sentence “Besides its role in cancers, other biological roles of Alk, such as involvement in the development and function of the nervous system, have also been discovered recently [2-4].", we presented recent findings in roles of Alk signaling and olanzapine-induced metabolic disorders in Introduction, as follows:

“Previous studies suggested that ALK is an important molecular mediator on the excitatory synaptic transmission onto nucleus accumbens shell medium spiny neurons expressed dopamine D1 receptors and ethanol consumption, suggesting ALK as one important molecular mediator of this interaction, and a potential target for therapeutic treatment of alcohol use disorder [3].” (see Page 1, lines 38-43)

“In particular, recent genome-wide association studies (GWAS) have suggested that several Alk variants are associated with metabolic traits such as adiponectin concentration [5], lipid and glucose homeostasis [6], and body mass index (BMI) [7, 8]. Recently, Orthofer et al. reported that the sympathetic control of adipose tissue lipolysis was linked to the Alk pathway modulating energy expenditure [9]. This finding came under the spot-light [10], and was supported by a recent study which found that Augmentor α (Augα), one of newly discovered ligands of Alk, appeared to control body weight through a hypothalamic pathway [11]. Since the sympathetic system and the parasympathetic system are the two arms of the autonomic nervous system [12]. It has been reported that specific mutations of the Alk gene induce an imbalanced development of the autonomic system, which leads to neuroblastoma in some patients [13]. Meanwhile, recent evidence links parasympathetic regulation to metabolic abnormalities such as abnormal food intake, weight gain, obesity and dyslipidemia [14-16]. It is rational to hypothesize that the parasympathetic system involves in ALK signaling mediated metabolic modulation.” (see Page 1, line 44 to Page 2, line 57)

“Second generation antipsychotics (SGAs), particular olanzapine, causes the highest incidence of metabolic side effects. Besides the increasing appetite stimulation and weight gain by blockade of hypothalamic 5HT2C and H1 receptors, it is a general consensus that the mechanisms underlying SGAs-induced metabolic disturbance are likely to be multi-factorial and involved in both peripheral and central mechanisms [23-27]. Our and other studies have identified that olanzapine has a high binding affinity to the M3R and can act as a potent M3R antagonists to directly induce dyslipidemia but the consequence of weight gain [28-30]. Co-treatment of cevimeline, a M3R agonist, can attenuate olanzapine-induced dyslipidaemia in animal model [31, 32]. These findings suggested that the ACh muscarinic M3R may be a prime mechanism of SGA-induced metabolic dysregulation, and a potential candidature of possible therapeutic targets. However, the underlying mechanism is largely unknown.” (see Page 2, lines 64-75)

  1. For such a small sample size, it is recommended to use non-parametric counterparts of statistical tests. This applies to both between-group differences and correlations. In this case, the applied parametric counterparts do not reflect the obtained results.

Reply: Thank you for the suggestion. Based on previous studies in our and other laboratories, the sample size has enough power detect the biological changes in metabolic parameters and mRNA expression in animals. If data is not normally distributed, we used non-parametric tests. If data is normally distributed, we used for example two-way ANOVA, which is suitable to effects of two factors and interactions between 2 factors.

  1. It is unclear in the results where and what test was used. It is recommended to report results according to standards, e.g., for the Mann-Whitney test: U = 23; p = 0.02.

Reply: Followed the comment, we added more details in statistical outcomes (including the statistical methods used) throughout the Results section.

  1. Appropriate effect size measures were not calculated for the applied statistical tests. The p-value alone is definitely not enough. Descriptive statistics should be expanded to include key measures, such as the median and 95% CI.

Reply: Followed the comments, we presented results in mean ± SD, or with median and 95% CI, throughout the Results section.

  1. The conclusions and discussion are not supported by a thoroughly conducted analysis. At this point, the article does not bring new findings.

Reply: Thanks for the comments. This paper provided novel evidence of a regulatory role of the M3R in Alk modulated lipid metabolism. These new findings have been analyzed and discussed from current literature relevant to the topic. (see Discussion section)

Reviewer 3 Report

Comments and Suggestions for Authors

Dear authors,

After reviewing this study, I have several concerns and suggestion to this study.

1. The rationale of this study is not clear. The authors did not disclose any information about the rationale and unmet medical need about olanzapine-induced dyslipidemia, and that will cause the readers unable to understand the main aims of this study.

2. Did the authors measure appetite of the model animal? It's well-known that olanzapine can reduce nausea and vomiting, and potentiate appetite in cancer patients administrated with advanced malignancies (10.1001/jamaoncol.2020.1052). Moreover, olanzapine shows potential in treating anorexia in clinics (10.1002/brb3.2498). How do the authors exclude over-diet-induced dyslipidemia in this study?

3. The authors need to disclose why you investigate alk and related signaling? Alk is not generally expressed in liver (https://www.proteinatlas.org/ENSG00000171094-ALK/tissue). How do the authors know alk is the susceptible target?

4. Please provide the histological staining of the desired proteins in this study. I agree that mRNA data is comparable to that of IHC. However, IHC data can show not only quantitative results but also spatial distributions of the target proteins which can not be observed in qPCR.

5. The dosing of olanzapine applied in this study need to be optimized. I agree that the dose applied in this study is effective based on the previous studies. However, the olanzapine-induced dyslipidemia in this study is ambiguous even though it's statistical significant. Moreover, the authors CHRM3 agonist to prove the effect of olanzapine-induced dyslipidemia is based on the reduction of CHRM3. However, the ambiduous improvement of olanzapine-induced dyslipidemia can not definitely display the offset of cevimeline.

6. The Materials and Methods is too brief, including the protocol of measuring and treatment schedule. I suggest the authors providing a schematic illustration about the treatment schedule.

7. Please provide the approval code of IACUC for this study. This is the mandatory information in in vivo study.

Comments on the Quality of English Language

I have no comment about the English writing. 

Author Response

  1. The rationale of this study is not clear. The authors did not disclose any information about the rationale and unmet medical need about olanzapine-induced dyslipidemia, and that will cause the readers unable to understand the main aims of this study.

Reply: Followed the comments. We have provided rationale and unmet medical need about olanzapine-induced dyslipidemia to justify the main aims of this study, as follows:

“Second generation antipsychotics (SGAs), particular olanzapine, causes the highest incidence of metabolic side effects. Besides the increasing appetite stimulation and weight gain by blockade of hypothalamic 5HT2C and H1 receptors, it is a general consensus that the mechanisms underlying SGAs-induced metabolic disturbance are likely to be multi-factorial and involved in both peripheral and central mechanisms [23-26]. Our and other studies have identified that olanzapine has a high binding affinity to the M3R and can act as a potent M3R antagonists to directly induce dyslipidemia but the consequence of weight gain [27-29]. Co-treatment of cevimeline, a M3R agonist, can attenuate olanzapine-induced dyslipidaemia in animal model [30, 31]. These findings suggested that the ACh muscarinic M3R may be a prime mechanism of SGA-induced metabolic dysregulation, and a potential candidature of possible therapeutic targets. However, the underlying mechanism is largely unknown. Therefore, this project aimed to further investigate whether Alk pathway is associated with Olanzapine induced metabolic disturbances via hepatic M3R mediated parasympathetic regulations.” (see Page 2, lines 64-77)

  1. Did the authors measure appetite of the model animal? It's well-known that olanzapine can reduce nausea and vomiting, and potentiate appetite in cancer patients administrated with advanced malignancies (10.1001/jamaoncol.2020.1052). Moreover, olanzapine shows potential in treating anorexia in clinics (10.1002/brb3.2498). How do the authors exclude over-diet-induced dyslipidemia in this study?

Reply: We agreed with the comments.  In previous studies, we have investigated the effects of olanzapine on the appetite and weight gain, as well as the underlying mechanisms via blockade of hypothalamic 5HT2C and H1 receptors. We have also found previously that olanzapine-induced dyslipidaemia is associated with weight gain. However, recent evidence suggested that SGAs could directly elevate fasting triglyceride levels without changes in body weight, because the mechanisms of SGA-induced metabolic disturbance are likely to be multi-factorial and involved in both peripheral and central mechanisms. In this study, we focused on the peripheral mechanisms of olanzapine-induced dyslipidaemia. We have made this clear in Introduction, as follows:

“Second generation antipsychotics (SGAs), particular olanzapine, causes the highest incidence of metabolic side effects. Besides the increasing appetite stimulation and weight gain by blockade of hypothalamic 5HT2C and H1 receptors, it is a general consensus that the mechanisms underlying SGAs-induced metabolic disturbance are likely to be mul-ti-factorial and involved in both peripheral and central mechanisms [23-27]. Our and other studies have identified that olanzapine has a high binding affinity to the M3R and can act as a potent M3R antagonists to directly induce dyslipidemia but the consequence of weight gain [28-30]….” (page 2, lines 64-71)

  1. The authors need to disclose why you investigate alk and related signaling? Alk is not generally expressed in liver (https://www.proteinatlas.org/ENSG00000171094-ALK/tissue). How do the authors know alk is the susceptible target?

Reply: Muscarinic acetylcholine M3 receptors are broadly expressed in the central nervous system and non-neural tissues. They are innervated by parasympathetic system [20,21]. The potential role of these receptors in the modulation of Olanzapine-induced activation of Alk signaling and related metabolic changes could not be completely excluded. We have made provided relevant information as follows:

“…Since the sympathetic system and the parasympathetic system are the two arms of the autonomic nervous system [12]. It has been reported that specific mutations of the Alk gene induce an imbalanced development of the autonomic system, which leads to neuroblastoma in some patients [13]. Meanwhile, recent evidence links parasympathetic regulation to metabolic abnormalities such as abnormal food intake, weight gain, obesity and dyslipidemia [14-16]. It is rational to hypothesize that the parasympathetic system in-volves in ALK signaling mediated metabolic modulation.” (Page 2, lines 51-57)

“Acetylcholine is the major neurotransmitter of the vagus nerves, which innervates the liver [17, 18]. It exerts parasympathetic actions by activating the acetylcholine (Ach) muscarinic M1 - M5 receptors (M1R-M5R), a group of heterotrimeric G protein-coupled receptors (GPCRs) [19]. Apart from the central nervous system, M3Rs are broadly expressed in the non-neural tissues including the hepatocytes [20, 21]. M3R plays a key role in regulating insulin secretion and other metabolic functions [22].” (Pag 2, lines 58-63)

“…Therefore, this project aimed to further investigate whether Alk pathway is associated with Olanzapine modulated Olanzapine induced metabolic disturbances via hepatic M3R mediated parasympathetic regulations.” (Page 2, lines 75-77)

  1. Please provide the histological staining of the desired proteins in this study. I agree that mRNA data is comparable to that of IHC. However, IHC data can show not only quantitative results but also spatial distributions of the target proteins which can not be observed in qPCR.

Reply: Thanks for the comment. However, this study examined only mRNA expression, but not IHC histological staining. We have discussed this limitation, as follows:

“One limitation is that only mRNA expression was examined in this study. Further studies are necessary to examine the protein levels with Western blot and Immunohistochemistry to fully reveal the regulation of parasympathetic M3R on Alk modulated lipid metabolism.” (see Page 10, lines 341-344)

  1. The dosing of olanzapine applied in this study need to be optimized. I agree that the dose applied in this study is effective based on the previous studies. However, the olanzapine-induced dyslipidemia in this study is ambiguous even though it's statistical significant. Moreover, the authors CHRM3 agonist to prove the effect of olanzapine-induced dyslipidemia is based on the reduction of CHRM3. However, the ambiduous improvement of olanzapine-induced dyslipidemia can not definitely display the offset of cevimeline.

Reply: Thank you for the comments.  We have discussed in more details to address these issues, as follows:

“Although the dosage of Olanzapine was 6 mg/kg/day in both Experiments 1 and 2, the daily dosage was delivered in two times (3 mg/kg by 2 times) in Experiment 1, but three times (2 mg/kg by 3 times) in Experiment 2. It has been reported that both 2 mg/kg and 3 mg/kg Olanzapine in rats have a clinically comparable dopamine D2 receptor occupancy (of approximately 65–80%)[47, 48]. In consideration of the half-life of Olanzapine is 24.2 and 72 hours respectively in plasma and brain of humans [49], while 2.5 and 5.1 hours in the plasma and brain (maintained at a high level for 8 hours) of rats [50], the Olanzapine dosage in Experiments 1 and 2 should be equivalent and has a clinically comparable D2 receptor occupancy. The effects of only one dosage of Cevimeline have been examined in this study, although these dosages have been proven to be effective in modulating lipid metabolism [31]. Further investigation with multiple dosages to confirm this finding is necessary. According to the dosage translation between species based on body surface area [51, 52], the Cevimeline dosage used in this study may be higher than the clinical dosage [32, 53]. However, the half-life of Cevimeline is shorter in rats (~2 hours) than in humans (~5 ± 1 hours) [53]. The current dosage has been reported without toxic effects in chronic studies up to 1 year, and without adverse effects on general behaviors, or nervous and other systems, although it may also act on other organs expressed muscarinic M3Rs [54-56].” (see Page 9, lines 323-340).

“…More importantly, hepatic Alk mRNA expression was significantly activated by Olanzapine in rats, while the activated Alk signaling was inhibited when stimulated with an M3R agonist Cevimeline. It is worth noting that, although Olanzapine is a potent M3R antagonist, it also binds to other neurotransmission receptors, such as dopamine D2, 5-HT2C, and histamine H1 receptors [42]. The potential role of these receptors in the modulation of Olanzapine-induced activation of Alk signaling and related metabolic changes could not be completely excluded. However, Cevimeline is an M3R agonist with a high degree of M3R specificity and only a low M1 receptor affinity [34]. Since both Alk signaling activation and metabolic disturbance induced by Olanzapine were found to be attenuated after co-treated with the Cevimeline in this study, this strongly supports the proposition that M3R regulates Alk-mediated metabolic traits.” (see Page 9, lines 294-304)

  1. The Materials and Methods is too brief, including the protocol of measuring and treatment schedule. I suggest the authors providing a schematic illustration about the treatment schedule.

Reply: Followed the comment. We added Supplementary Figure 1 to illustrate the animal treatment schedule in our study.

  1. Please provide the approval code of IACUC for this study. This is the mandatory information in in vivo study.

Reply: We have added the code of IACUC by Animal Ethics Committee from University of Wollongong, as follows:

“All of the experimental processes were approved by the Animal Ethics Committee, University of Wollongong (AE12/26 & AE15/03), complying with the Australian Code of Practice for the Care and Use of Animals for Scientific purposes complying with the Australian Code of Practice for the Care and Use of Animals for Scientific Purposes.” (see Page 2, lines 87-91)

Reviewer 4 Report

Comments and Suggestions for Authors

This is a straightforward study in adult female rats of the effects of chronic oral dosing with a single dose of olanzapine, a second generation atypical antipsychotic, upon markers of lipid metabolism, adiposity and body weight (BMI). The authors also probed with oral dosing of a single M3 muscarinic cholinergic receptor agonist, by itself or in combination with olanzapine dosing, on these same parameters. The authors report body weight, blood assays for cholesterol and lipids, and liver mRNA levels (assayed by RT-qPCR) for anaplastic lymphoma kinase (ALK) gene and other ALK regulatory genes. They report that olanzapine treatment increased BMI and adiposity indices, down-regulated liver ALK expression and its gene promoters, upregulated ALK inhibitor. These changes were partially reversed by M3 agonist co-treatment.

Were the drug treatments with olanzapine and M3 agonist also 9 weeks in duration? This does not appear to have been stated.

The authors conclude that olanzapine can facilitate weight gain and adiposity by these mechanisms that are mediated through the M3 muscarinic receptor system. This seems like a reasonable conclusion given their data, but there are concerns about how they performed the experiments and analyzed the data.

First, as the authors point out, they studied only a single dose of olanzapine and a single dose of M3 agonist. Second, and not discussed, is why they studied only female rats (ie, no males). In some journals, this gender restriction in a clinical drug study would lead to automatic rejection, unless a compelling reason is presented for studying a single gender.

They eliminate outlier data using box plots. If I am counting correctly, this outlier rejection affected only the "olanzapine" group, as all other groups showed 12 data points for each bar. Is this correct? Do the authors have any explanation for this outlier?

By the way, the graphs should show mean +/- S.D. (not SEM). This is an easy change to make in the graphics software they used.

The correlation graphs don't add any mechanistic data and could easily be presented as Supplemental Data.

So this is an interesting paper as far as olanzapine goes. As the authors know, other antipsychotic drugs are used clinically, and many cause weight gain. Might they also act through a M3 muscarinic receptor mechanism?

I suggest that the authors address the above points. If they cannot justify use of female rats only, then that is a serious deficit that could lead to rejection. I'll await their revision, if they choose to make one.

Comments on the Quality of English Language

Overall English is fine, with rare subject-verb corrections that are easily remedied.

Author Response

  1. Were the drug treatments with olanzapine and M3 agonist also 9 weeks in duration? This does not appear to have been stated.

Reply: Yes, it is correct that then treatment with olanzapine and M3 agonist was also 9 weeks in Experiment 2.  This was presented in methods (see Page 3, line 106) and Supplementary Figure 1.

  1. The authors conclude that olanzapine can facilitate weight gain and adiposity by these mechanisms that are mediated through the M3 muscarinic receptor system. This seems like a reasonable conclusion given their data, but there are concerns about how they performed the experiments and analyzed the data.

First, as the authors point out, they studied only a single dose of olanzapine and a single dose of M3 agonist. Second, and not discussed, is why they studied only female rats (ie, no males). In some journals, this gender restriction in a clinical drug study would lead to automatic rejection, unless a compelling reason is presented for studying a single gender.

Reply: We agreed with the Reviewer. We have discussed these limitations as follows:

“It is worth noting there are several limitations in this study. One limitation is that only mRNA expression was examined in this study. Further studies are necessary to examine the protein levels with Western blot and Immunohistochemistry to fully reveal the regulation of parasympathetic M3R on Alk modulated lipid metabolism. In addition, it is also valuable in a future study to investigate the effects of M3R knockout on Alk-modulated lipid metabolism. In addition, only female rats were used in this study, because the female patients were observed to more sensitive to Olanzapine-induced metabolic disturbances in both humans and rats [57-59]. Indeed, a rodent model of Olanzapine induced metabolic disturbance could not be consistently constructed unless given a higher fat diet instead of the standard low-fat chow [60]. This limitation would hinder our results to be transferred to a clinical setting, therefore further research is necessary to examine the effects of cevimeline intervention in preventing metabolic side effects of Olanzapine in a male cohort.” (see page 10, lines 341-353)

  1. They eliminate outlier data using box plots. If I am counting correctly, this outlier rejection affected only the "olanzapine" group, as all other groups showed 12 data points for each bar. Is this correct? Do the authors have any explanation for this outlier?

Reply: Actually, one animal in olanzapinde group accidentally died during the experiment period in Experiment 1. Therefore, there was only 11 data in this group in Figure 1. We have explained it in Supplementary Figure 1.

  1. By the way, the graphs should show mean +/- S.D. (not SEM). This is an easy change to make in the graphics software they used.

Reply:  Followed the comment, we have changed presentations from mean +/-  SEM to mean +/- SD in all the graphs.

  1. The correlation graphs don't add any mechanistic data and could easily be presented as Supplemental Data.

Reply:  Followed the comments, we have revised all correlation graphs and presented them as Supplementary Figure 2 and Supplementary Figure 3.

  1. So this is an interesting paper as far as olanzapine goes. As the authors know, other antipsychotic drugs are used clinically, and many cause weight gain. Might they also act through a M3 muscarinic receptor mechanism?

Reply:  Yes, besides Olanzapine, antipsychotic drug Clozapine may also cause metabolic disorders through a M3 muscarinic receptor mechanisms. Now I discuss this issue, as follows:

“Besides Olanzapine, it has been reported that Clozapine also induced metabolic disorders through a muscarinic M3 receptor mechanism [30]. Further studies are important to investigate whether Clozapine-induced metabolic disorders are also mediated by Alk signalling, and could be attenuated by co-treated with Cevimeline. “  (see Page 10, lines 361-365)

  1. I suggest that the authors address the above points. If they cannot justify use of female rats only, then that is a serious deficit that could lead to rejection. I'll await their revision, if they choose to make one.

Reply: Followed the comments, we have discussed the limitation and justified why only female rats were used in this study, as follows:

“In addition, only female rats were used in this study, because the female patients were observed to more sensitive to Olanzapine-induced metabolic disturbances in both humans and rats [57-59]. Indeed, a rodent model of Olanzapine induced metabolic disturbance could not be consistently constructed unless given a higher fat diet instead of the standard low-fat chow [60]. This limitation would hinder our results to be transferred to a clinical setting, therefore further research is necessary to examine the effects of cevimeline intervention in preventing metabolic side effects of Olanzapine in a male cohort.” (see Page 10, lines 346-353)

Round 2

Reviewer 2 Report

Comments and Suggestions for Authors

The article has been revised; however, the statistical analysis still requires several changes. Not all recommendations have been taken into account. For instance, the appropriate effect size measures have not been calculated. The discussion and conclusions are not based on a properly conducted analysis (inappropriate tests, etc.).

Comments on the Quality of English Language

Minor editing of English language required.

Author Response

Comments and Suggestions for Authors

The article has been revised; however, the statistical analysis still requires several changes. Not all recommendations have been taken into account. For instance, the appropriate effect size measures have not been calculated. The discussion and conclusions are not based on a properly conducted analysis (inappropriate tests, etc.).

Reply: Many thanks for the comments. Now we have made clear how sample sizes were calculated in this study. Our previous studies have confirmed that the sample sizes used in this study have enough power to detect the changes induced by antipsychotic drugs. As follows:

“Based on the data from our previous studies, a power analysis (with type I error of 0.05 and 80% power) was conducted to determine sample sizes [34-37]. It has been confirmed in our previous studies that sample sizes (n=12/group for metabolic parameters and n=6/group for mRNA expressions) have an enough power to detect the changes induced by antipsychotic intervention in rats [34, 36-39].” (see Section “2.1 Animal experimental design”, lines 100-105)

Regarding the statistical tests, we have already presented clearly how data distributions were analyzed. Based on data distribution, relevant parameters and non-parametric tests were used. (see section “2.5. Statistical analysis”, lines 160-173) These statistical methods have been used successfully to investigate antipsychotic induced metabolic disorders and underlying mechanisms in our previous studies [references 34, 36-39]. Therefore, we have made the discussion and conclusions properly based the analysis in this study.

We have also checked English throughout the manuscript.

Reviewer 3 Report

Comments and Suggestions for Authors

Thank you for your reply. I have no further concern regarding to this study.

Author Response

Many thanks for the positive comment.

Reviewer 4 Report

Comments and Suggestions for Authors

The authors have extensively revised this paper along the guidelines I provided in my review of the original manuscript. Although there remain limitations, they have acknowledged them. In particular, they provide a reasonable explanation for the exclusive use of female rats (as opposed to both genders). They also note that only one daily dose of olanzapine was used, and additional research will be needed to discover any dose-response variation. With this additional information, it is possible to envision a clinical study emerging from their preclinical research.

Comments on the Quality of English Language

Overall English is fine. There are a very few subject-verb mismatches, but these are easily corrected with careful editing.

Author Response

Reply: Many thanks for positive comments.

We have checked the whole manuscript and corrected the subject-verb mismatches.

Round 3

Reviewer 2 Report

Comments and Suggestions for Authors

Unfortunately, the recommended statistical guidelines have still not been accepted. The results are not supported by the application of appropriate statistical tests.

Comments on the Quality of English Language

Minor editing of English language required.

Author Response

The reviewer’s comments are appreciated. Since the reviewer did not provide specific comments (in statistical tests) in Round 2 review, we have read again the reviewer’s specific comments in original and revised manuscripts. In Revision 1 and Revision 2, we have addressed these specific comments. One issue that the reviewer criticized is the sample size. In comparison with clinical trials normally with a large sample size, the Animal Ethics guidelines in 3 R’s (Replacement, Reduction, Refinement) request “reduction of animals”, therefore animal tests are normally conducted in small sample size. We have followed the statistical guidelines to determine sample sizes by a power analysis, and to conduct following data analysis. We also followed the ARRIVE guidelines (Animal Research: Reporting of In Vivo Experiments) to report statistical analysis and results. Therefore, the results are solid and appropriate.

We agreed there is a limitation of a relatively small sample sizes, in comparison with clinical trials. Now we discussed the limitation as follows:

“Another limitation is that the sample size is relatively small, although a power analysis has been conducted based on the data from our previous studies. There-fore, further experiments in a large sample size will be valuable to confirm the findings in this study.” (see line 432-435)

We have rechecked the manuscript and made some minor corrections.